# Impact of Estrogen on Purinergic Signaling in Microvascular Disease

**DOI:** 10.3390/ijms26052105

**Published:** 2025-02-27

**Authors:** Jessica Cassavaugh, Maria Serena Longhi, Simon C. Robson

**Affiliations:** Department of Anesthesia, Critical Care and Pain Medicine, Beth Israel Deaconess Medical Center, Boston, MA 02215, USA; mlonghi@bidmc.harvard.edu (M.S.L.); srobson@bidmc.harvard.edu (S.C.R.)

**Keywords:** estrogen, microvascular, purinergic, adenosine, ATP, cardiovascular

## Abstract

Microvascular ischemia, especially in the heart and kidneys, is associated with inflammation and metabolic perturbation, resulting in cellular dysfunction and end-organ failure. Heightened production of adenosine from extracellular nucleotides released in response to inflammation results in protective effects, inclusive of adaptations to hypoxia, endothelial cell nitric oxide release with the regulation of vascular tone, and inhibition of platelet aggregation. Purinergic signaling is modulated by ectonucleoside triphosphate diphosphohydrolase-1 (NTPDase1)/CD39, which is the dominant factor dictating vascular metabolism of extracellular ATP to adenosine throughout the cardiovascular tissues. Excess levels of extracellular purine metabolites, however, have been associated with metabolic and cardiovascular diseases. Physiological estrogen signaling is anti-inflammatory with vascular protective effects, but pharmacological replacement use in transgender and postmenopausal individuals is associated with thrombosis and other side effects. Crucially, the loss of this important sex hormone following menopause or with gender reassignment is associated with worsened pro-inflammatory states linked to increased oxidative stress, myocardial fibrosis, and, ultimately, diastolic dysfunction, also known as Yentl syndrome. While there is a growing body of knowledge on distinctive purinergic or estrogen signaling and endothelial health, much less is known about the relationships between the two signaling pathways. Continued studies of the interactions between these pathways will allow further insight into future therapeutic targets to improve the cardiovascular health of aging women without imparting deleterious side effects.

## 1. Introduction

Microvascular disease (MVD) is a process of small vessel dysfunction within arterioles, venules, and capillaries [1]. In contrast to the well-studied, large vessel atherosclerotic diseases, coronary MVD was originally believed to be associated with a lower risk of major adverse cardiac events. More recently, however, coronary MVD has been correlated with a much higher than originally believed incidence of major cardiac events, including heart failure, stroke, and death [2]. Multiple factors contribute to the development of microvascular disease. The traditional macrovascular disease risk factors such as tobacco, hyperlipidemia, obesity, and diabetes mellitus, are also significant risk factors for the development of MVD. Other risk factors include systemic processes that alter tissue perfusion, including micro-thromboembolism or arterial hypertension [3]. While MVD may occur in any microcirculatory bed (peripheral vasculature, brain, kidney, etc.), coronary microvessel dysfunction is of specific interest as it results in deranged tissue oxygenation, inflammation, and ischemic heart disease, with its associated morbidity and mortality. 

Microvascular disease, that occurs at a disproportionally higher rate in females, also known as Yentl syndrome, remains more difficult to diagnose and has fewer treatments available than macrovascular disease [4]. The Women’s Ischemia Syndrome Evaluation (WISE) study recently assessed 10-year mortality rates and observed that, out of the group of women with signs and/or symptoms of ischemia, one in three deaths occurred in those without obstructive coronary disease. Not surprisingly, mortality rates remain significantly higher in females with microvascular disease when compared to healthy controls [5].

Estrogens are known to have a protective effect on the cardiovascular system, principally in small vessels [6]. The gender-specific discordance in MVD is, at least in part, due to the loss of estrogen that occurs with aging. However, simple hormone replacement (HRT) is not beneficial and may increase the risk of thrombotic adverse events. Two large, randomized control trials, the Women’s Health Initiative (WHI) and the Heart and Estrogen/progestin Replacement Study (HERS), failed to demonstrate HRT-associated prevention of cardiovascular events, including death, myocardial infarction, or angina [7]. HRT is associated with increased stroke and pulmonary embolism, although this risk is variable depending on specific patient characteristics. Because of the challenges of treating vasomotor symptoms of menopause, newer HRT trials are focused on the specific timing of the initiation and route of administration to design a safer approach for patients. Data for non-menopausal patients’ use of exogenous estrogen is even sparser and largely extrapolated from prior trials of HRT in females, including the aforementioned WHI and HERS, as well as from oral contraception adverse events data [8,9]. 

Commonly occurring under ischemic conditions, acute inflammation is a major contributor to the development and ongoing propagation of microvascular disease. The cellular response to microvascular injury is multifaceted and involves the release of inflammatory mediators, vasodilators, platelet activation, and reactive oxygen species. Nucleotides are widely known to be active signaling molecules and have been shown to directly modulate platelet, endothelial, and leukocyte responses [10]. Pivotal work originating from Geoffrey Burnstock demonstrated that adenosine 5′-triphosphate (ATP) release from endothelial cells occurs under inflammatory conditions and shear stress [11,12]. Various other cell types, including myocytes, erythrocytes, and neurons, have also been shown to release ATP under hypoxic conditions [13,14,15]. Close regulation of ATP through hydrolysis to adenosine via the purinergic signaling pathway enzymes is necessary to maintain a balance between pro- and anti-inflammatory and thrombotic responses. The interactions between ATP, adenosine, and their receptors are essential to preserving endothelial homeostasis in the extracellular milieu. 

In this review, the relationship between MVD, purinergic signaling, and estrogens will be explored. A particular focus will be placed on the impacts on endothelial health of both estrogen and purinergic responses. 

## 2. Purinergic Signaling

First proposed in 1972, purinergic signaling has rapidly been identified as a key regulatory pathway for homeostasis in neuronal, endothelial, and immunological systems. ATP and adenosine are responsible for regulating many intracellular functions, including the release of signaling molecules (neurotransmitters, chemokines, etc.), smooth muscle contraction, and metabolic functions. The activities of extracellular ATP and adenosine on their various receptor subtypes (P2 and P1, respectively) is an ongoing topic of interest, especially as their effects vary depending on the receptor subtype, the cell type, and the environmental input [16]. For example, healthy endothelial activation of the P2Y receptor, an ATP receptor, inhibits platelet aggregation. However, platelet aggregation is also activated via adenosine diphosphate (ADP) operating via P2X1, P2Y_1,_ and P2Y_12_ in response to endothelial and vascular signals—von Willebrand factor, collagen, and others. For an in-depth review of purinergic receptors and their effects, please see PMID: 28057794. 

ATP and adenosine concentrations are tightly regulated through interactions with cell membrane ectoenzymes. Specifically, two ectonucleotidases, CD39 and CD73, are responsible for maintaining a balance between ATP and adenosine. The ecto-nucleoside triphosphate diphosphohydrolase (ENTPDase) family of enzymes, of which there are several (1, 2, 3, and 8), hydrolyze 5′-triphosphates. Specific to the vasculature, CD39 (ectonucleoside triphosphate diphosphohydrolase-1, (ENTPD1, NTPDase1)) catalyzes the hydrolysis of the phosphoanhydride bonds of ATP and ADP to adenosine monophosphate (AMP) [17,18,19]. Hydrolysis of AMP is then catalyzed via CD73 (ecto-5′-nucleotidase (NT5E)) to the primarily anti-inflammatory molecule, adenosine [18]. The ectonucleotidase signaling cascade is rate-limited by CD39 activity, with AMP hydrolysis through CD73 occurring very rapidly. Homeostatic regulation of these ectoenzymes is essential to the health of various tissues, including the endothelium.

CD39 and CD73 are present in a variety of cell types, including endothelial, smooth muscle, dendritic, and lymphocytes. Overexpression of CD39 leads to reduced thrombus formation through attenuation of platelet activation via decreased integrin GPαIIb/β3 activity and offers protection from myocardial ischemia in vivo [20,21,22]. Conversely, the knock-down of CD39 results in a pro-inflammatory phenotype characterized by endothelial, hemostatic, and leukocyte dysfunction [23,24]. CD73 regulation of adenosine production maintains vascular barrier integrity during hypoxia, while CD73-deficient mice are more susceptible to neointimal plaque formation and macrophage accumulation [25,26]. Both CD39 and CD73 are shown to be transcriptionally regulated by hypoxia, thus suggesting the critical importance of regulation of nucleotides in the modulation of vascular health and integrity [27,28]. 

### 2.1. Purines

#### 2.1.1. Adenosine

Adenosine, a ribonucleoside made up of adenine and ribose, was first identified in 1929 [29]. Adenosine functions as an essential modulator of cardiovascular health, largely through coronary vasodilation and anti-inflammatory effects [30]. The ratios of ATP to adenosine and enzymatic activity are highly specific to the cell type in question. In endothelial cells, as demonstrated in essential work by Yegutkin et al., a high level of ATP/ADP hydrolysis exists that results in an adenosinergic dominant environment. This activity is due in part to high NTPDase and ecto-5′-nucleotidase activity and is believed to support an anti-inflammatory, anti-thrombotic milieu. In contrast, other cell types, such as non-regulatory or helper-type lymphocytes, promote an ATP-generating/adenosine-eliminating environment [31]. 

Intracellular levels of adenosine are tightly controlled through various mechanisms, including transmembrane shuttling, recycling, and rapid deamination (Figure 1). Two transmembrane channels, concentrative nucleoside transporters (CNT) and equilibrative nucleoside transporters (ENT), allow for cross-membrane adenosine movement. Both channels allow for bidirectional adenosine movement in either a sodium-dependent (CNT) or -independent (ENT) manner [32]. ENTs are highly expressed throughout most tissues and within the vasculature, whereas the CNT family of transporters are more commonly found in the specialized epithelium, including renal and intestinal, making the ENT class of transporters of interest for purine regulation in the vasculature (see the section on Purine Modulation below). 

Adenosine removal is carried out through several mechanisms. Adenosine deamination is responsible for approximately 90% of adenosine degradation and is carried out through the deamination of adenosine to inosine via the enzymatic activity of adenosine deaminase (ADA). Inosine is either metabolized to uric acid or is recycled through the purine salvage pathway into purine precursors. A second major adenosine removal pathway is through adenosine kinase (ADK) recycling of adenosine to ATP through re-phosphorylation [33,34,35,36]. 

Regulation of adenosine production occurs through several tightly modulated intra- and extracellular pathways. The first is through ATP degradation pathways, which occur both at the cell membrane and intracellularly, albeit with differing enzymes. Extracellular ATP is hydrolyzed through the ENTPDases, while in the cytosol, ATP is de-phosphorylated to ADP by a nucleoside diphosphate kinase. The next de-phosphorylation step, catalyzed by adenylate kinase, results in AMP, which is subsequently transformed to adenosine by 5′-nucleotidase [34,36,37]. A separate method of adenosine production is through the cyclic AMP-adenosine (cAMP) pathway. Briefly, cAMP, via a phosphodiesterase, is converted to AMP, which is then de-phosphorylated to adenosine by 5′-nucleotidases [38,39,40]. Adenosine can also be generated through hydrolysis of S-adenosyl homocysteine by S-adenosyl homocysteine hydrolase [41,42]. 

Extracellular adenosine activates adenosine receptors on the cell surface. These P1 class, G-protein coupled receptors, of which there are four subtypes, are expressed across multiple tissue types, with the A_2A_ and A_2B_ receptors the predominant types found on endothelial cells (Figure 2). The A1 adenosine receptor plays a role in vascular homeostasis, with prior studies suggesting that activation of A1 adenosine receptors stimulates vascular smooth muscle contraction through protein lipase C signaling. This negatively modulates vascular relaxation, mediated by other adenosine receptor subtypes (A_2A_) [43,44]. Both the A_2A_ and A_2B_ subtypes activate cyclic AMP, leading to mitogen-activated protein kinase (MAPK) activation. The resultant microvascular protective effects arise from increases in angiogenesis and the production of endothelial nitric oxide. Their effects also include inhibition of pro-inflammatory responses as well as fibroblast and smooth muscle growth [33,45,46]. Moreover, adenosine prevents pathogenic remodeling by reducing the release of norepinephrine, endothelin, and tumor necrosis factor-alpha (TNFα), thus preventing ongoing vascular pathology [47]. Due to the protective effects of adenosine on cardiovascular health and in some cancers, emerging targets of adenosine modulation by ADK and ADA inhibitors are currently under investigation [36,48,49].

#### 2.1.2. Adenosine 5′-Triphosphate

ATP is well known as the primary energy source of the cell. Synthesized by the electron transport chain, ATP is used in a variety of intracellular processes, including myocyte contraction, phosphorylation of proteins, and as a cofactor for membrane transport. ATP is also known to be an extracellular signaling molecule and, over the past several decades, has emerged as a significant actor in various tissue types, including neuronal, vascular, and immunological. Through interactions with its various P2 receptor subtypes, ATP regulates many signal transduction pathways, including neurotransmission and hormone secretion, as well as regulation of solid organs, platelets, and immunological function. 

Intracellular ATP is maintained in the 1–10 mM range, while extracellular ATP (eATP) is approximately 1000-fold lower at around 10 nM [50]. This high concentration gradient allows for a rapid increase in activity when released and prevents excessive P2 receptor stimulation. The high intracellular ATP concentration is also needed to maintain the ATP/ADP ratio that drives metabolic reactions and to keep proteins soluble through hydrotrope functionality [51]. ATP is produced through various intracellular metabolic cascades, including glycolysis, the tricarboxylic acid (TCA) cycle, and oxidative phosphorylation (Figure 3). Glycolysis produces very limited amounts of ATP, whereas the TCA cycle and electron transport chain generate greater than thirty molecules of ATP for every one of glucose. ATP is also produced within the cell through adenosine re-phosphorylation to ATP, a multistep, kinase-driven process (Figure 1). 

Controlled ATP release occurs through various mechanisms, including plasma membrane channels and transporters and exocytosis [52]. Two primary transmembrane transporters shuttle ATP into the extracellular space. Pannexin 1, a heptameric membrane channel, is relatively quiescent at baseline, but with stimulation by stretch, ATP receptor signaling, and caspase-mediated cleavage, a large flux of ATP has been measured [53]. Another family of ATP channels, connexins, opens in response to pathologic stimulation, including oxidative stress and changes in intracellular calcium concentrations. Pannexin and connexin activities, in combination with the accompanying ATP release, are shown to be associated with inflammation, vasoconstriction, and ischemic/reperfusion injury [54,55].

ATP is also released at times of cellular death; during apoptosis, it is released in a controlled manner as a “find me” signal for phagocytes. However, with pathologic necrosis that occurs during sudden cellular injury, the plasma membrane will rupture, and a large concentration of ATP will enter the extracellular milieu [56,57]. In addition, at times of ischemia, inflammation, or shear stress, cells, especially endothelial and erythrocytes, activate a controlled release of ATP into the blood vessel lumen. Once released, ATP acts as any other extracellular signaling molecule—interacting with its receptors, the inotropic P2Y and the metabotropic P2X.

ATP can be protective in the short-term response to injury through various mechanisms, including vasodilation by nitric oxide production. Additionally, following injury, a large burst of eATP can activate immunostimulatory pathways on neighboring immune cells and promote wound healing [58]. However, long-term signaling through ATP receptors results in pathologic activation of pro-inflammatory pathways such as the NOD-, LRR- and pyrin domain-containing protein 3 (NLRP3) inflammasome (Figure 4) [59,60]. As such, maintaining balance in the activation of ATP receptors with ATP hydrolysis to adenosine is crucial for cellular well-being.

### 2.2. Role of Purinergic Signaling in Microvascular Dysfunction

Purinergic pathway signaling is associated with many inflammatory conditions, including pulmonary fibrosis, inflammatory bowel disease, and neurodegenerative diseases, among others. In the cardiovascular system, purinergic signaling is especially important for maintaining homeostasis, the absence of which can lead to coronary microvascular dysfunction. From regulation of the autonomic nervous system to supporting endothelial integrity to promoting platelet function, ATP and adenosine have regulatory roles in nearly every cardiovascular cellular process. Purines have an immediate, local effect, with increases in the extracellular concentrations at times of injury or endothelial and/or platelet activation [61,62,63]. The vascular response to purine release is highly variable and contingent on the specific vascular bed, the cell type (endothelial versus smooth muscle versus neuron), and the receptor subtype.

#### 2.2.1. Inflammation

Microvascular dysfunction that occurs with chronic exposure to low-grade systemic inflammation is a significant contributor to MVD. This dysfunction appears to complicate common conditions such as obesity, diabetes, and other metabolic syndromes and is linked to the persistent generation of reactive oxygen species (ROS), the release of pro-inflammatory cytokines and, seemingly, disorders in purinergic pathways [64,65]. ROS and cytokines, such as TNFα, produce various types of tissue or cellular injury, which then activate a purine-mediated inflammatory response in the cardiovascular system [59,64].

Erythrocytes, immune, and endothelial cells do not possess the complete cellular enzymatic machinery under basal conditions for de novo purine synthesis. Hence, the cells of the blood and vasculature maintain intracellular ATP nucleotide pools by relying on salvage pathways, i.e., by taking up nucleoside derivatives, salvaging preformed nucleosides, and precursors imported from the extracellular milieu. The liver serves not only as the bioenergetic metabolic powerhouse of the body but also contributes in a major way to systemic purine synthesis by transferring metabolites (5-aminoimidazole-4-carboxamide ribonucleotide (AICAR) to AICAR transformylase) into erythrocytes/erythroblasts, and involving other synthetic pathways to maintain vascular homeostasis [66,67,68]. Importantly, hepatic tissues are estrogen-responsive, where the lack of this hormone profoundly alters cellular bioenergetics, peroxisome proliferator-activated receptor-gamma coactivator-1α function, and mitochondrial metabolism [69]. For this purpose, hepatocytes have a high density of mitochondria to support de novo purine synthesis, as well as to drive the various other metabolic functions, including detoxification, lipid metabolism, and gluconeogenesis, required for immunity and host defense [70]. 

Adenosine and the associated P1 receptors are largely protective from inflammatory processes in endothelial cells (Figure 2). Specifically, A_2A_ agonists prevented leukocyte migration and deposition into the vasculature, and A_2A_ receptor signaling inhibits proliferation and activation of T-cells [71,72]. In an in vivo mouse model of renal ischemia, the A1 adenosine receptor also served a protective function against inflammatory injury through decreases in pro-inflammatory cytokine release [73]. Adenosine receptor expression on endothelial cells is upregulated in the presence of inflammatory cytokines, and adenosine also promotes anti-inflammatory cytokine production, including interleukin 10 (IL-10). Ongoing inflammation is also prevented through the acceleration of tissue repair and angiogenesis, primarily through regulation of vascular endothelial growth factor (VEGF) and wound healing rates [74,75,76].

ATP-mediated signaling pathways are highly characterized in the endothelial inflammatory response. Extracellular ATP is released through a variety of mechanisms, including cell damage, lipid exposure, and oxidative stress. Specifically, inhibition of P2Y decreases monocyte invasion, as measured using an endothelial cell-specific knock-out mouse [77]. Endothelial P2X and P2Y receptors have also been implicated in both the activation of inflammasome and in nuclear factor-kappa B (NF-κB) activity (Figure 4). ATP stimulation activates the P2 receptor at the cell membrane, triggering calcium release, which then activates the NLRP3 complex. The P2 receptors also stimulate mitochondrial ROS production with subsequent NF-κB activation. Combined, these pathways increase the production of pro-inflammatory cytokines IL-6, and IL-1β, alter nitric oxide production, and support continued ROS production [78,79].

#### 2.2.2. Hypoxia

Tissue hypoxia, the imbalance of oxygen supply and demand, can occur following reduced arterial delivery of oxygen or following increased oxygen consumption, as frequently occurs with inflammation. Acute adaptations occur either directly through ATP receptor interactions or through increased adenosine concentration. Hypoxia directly triggers ATP release from erythrocytes and endothelial cells into the blood vessel lumen. ATP acting through its P2 receptors on the luminal side of the vessel can stimulate vasodilation through nitric oxide release [80]. 

Various proteins in the purinergic pathway are tightly regulated during hypoxia, with the overall response of increasing vasodilation through increased adenosine production. ATP hydrolysis is amplified through augmented regulation of the cell membrane ectoenzymes CD39 and CD73. Transcription of CD39 and CD73 is regulated by the hypoxia-induced transcription factors specificity protein 1 and hypoxia-inducible factor 1-alpha, respectively [27,28]. This increase in ectonucleotidase expression allows the microenvironment to cope with elevated eATP, as is seen with acute hypoxia or inflammation.

Adenosine, largely acting as an adaptive signaling molecule, is highly regulated in hypoxia. Adenosine receptors are known to be upregulated in hypoxia [81]. Furthermore, intracellular adenosine transport is reduced through reduction in ENT1, the cell membrane transport channel [82]. Lastly, adenosine degradation is reduced through the inhibition of adenosine kinase activity in hypoxia [83]. Combined, these hypoxia-dependent protective mechanisms result in endothelial vasodilation through additional extracellular adenosine.

#### 2.2.3. Shear Stress

The friction generated by blood flow on the apical surface of a vessel is known as shear stress. Alterations in shear stress that occur with stenotic flow or vascular stiffening result in turbulent flow, vascular remodeling, and a pro-inflammatory phenotype [84,85]. The normal physiological response to changes in flow includes ATP release from smooth muscle cells, endothelial cells, and sympathetic neurons [86]. Shear stress in the endothelium induces ATP release that mediates a mechanotransduction response through P2 receptor-mediated calcium release [12,87]. This ATP release can be both protective and pathological, depending on the duration and specific receptor activation. 

The vasodilatory effect through regulation of endothelial nitric oxide (eNOS) production is stimulated through multiple purinergic receptors. The production and release of eNOS in coronary arterioles were shown to be a direct effect of the A_2A_ adenosine receptor activity [88]. Interactions between ATP and its P2 receptors on endothelial cells also lead to nitric oxide release and vasodilation. Gonçalves da Silva et al. established the phosphorylation of eNOS to be mediated through P2Y-dependent signaling [89]. Similarly, recent work by Favre et al. has demonstrated CD39 regulation of ATP-dependent relaxation in the endothelium, at least partially through P2 receptor regulation [80,90]. 

While many of the purinergic responses to changes in shear stress are adaptive and protective, prolonged changes in flow or damage to the endothelium, however, lead to disrupted vasodilatory responses. For example, activated P2X ATP receptors on the vascular smooth muscle will stimulate pathological vasospasm through poorly coordinated contraction [87,91]. Endothelium that is continuously exposed to shear stress demonstrates changes in proliferation, migration, and integrin production [92]. Furthermore, dysregulated shear stress-induced ATP release can also trigger endothelial dysfunction, pro-inflammatory signaling through transcription factor upregulation, and activation of the NLRP3 inflammasome [78,93,94]. 

#### 2.2.4. Thrombosis

Healthy endothelium maintains a quiescent anti-thrombotic state, while injured glycocalyx and/or endothelium can rapidly induce platelet and coagulation pathway activation, resulting in blood clots. Microvascular blood clots form not only from dysregulated coagulation but also from dysregulated inflammation, a condition referred to as thromboinflammation. During times of high shear stress or acute inflammation, platelet aggregation and adhesion occur. Inflammation activates the endothelium, and it responds by releasing platelet activation molecules such as von Willebrand factor and selectins, as well as platelet-mediated granulocyte release of leukocyte stimulatory cytokines; these cytokines (IL-8, platelet factor 4, etc.) activate circulating neutrophils, leading to the release of fibrin by platelets [95,96]. Furthermore, endothelial cell activation also suppresses anti-coagulation by the loss of surface heparin and thrombomodulin [97]. Additionally, endothelial injury, such as may occur with significant shear stresses or ROS exposure, uncovers tissue factor, which stimulates activation of the coagulation cascade [62]. The overall response results in an ongoing cascade of platelet activation and disordered clot formation. 

Under conditions of chronic inflammation, levels of the constitutively expressed purines are significantly increased. The rapid hydrolysis of ADP to adenosine by CD39 is critical for preventing inappropriate ADP-mediated platelet activation through ADP receptors [62]. Previous studies of cerebral microvasculature demonstrated CD39 to be essential in maintaining a normal thrombotic phenotype, as the CD39−/− animals revealed a prothrombotic phenotype following an ischemic injury [98]. However, the hydrolysis of ATP/ADP to adenosine in the microenvironment is not rapid enough to prevent local P2X1 receptor activation [99]. Adenosine receptors (A_2A_, A_2B_) on platelets produce the opposite effect—through adenosine-mediated protein kinase A (PKA) and C activation, platelet activity and adherence are inhibited, as demonstrated in both in vitro and in vivo studies. A_2A_ receptors have also been shown to inhibit the actions of thrombin, including barrier functions and thrombin-mediated pro-inflammatory cytokine activation [100,101]. 

## 3. Estrogen Signaling

Estrogens are a group of steroid hormones known to regulate growth and differentiation in a variety of tissues. Three estrogens make up the group of hormones: the predominant and most potent estrogen, 17β-estradiol (E2), as well as estrone (E1) and estriol (E3) [102,103,104]. Estrogens are not only essential to reproductive health, but they also have physiologic roles in nearly every other organ system. Their effects are mediated through two types of receptors—the steroid hormone nuclear receptors, estrogen receptor alpha and beta (ERα, ERβ), and the G-protein coupled receptor, G protein-coupled estrogen receptor 1 (GPER) [105,106,107,108]. 

The nuclear receptors, ERα and ERβ, function as ligand-activated transcription factors and mediate the genomic pathway of estrogen signaling. Upon binding estrogen, these receptors dimerize, translocate to the nucleus, and direct the transcription of target genes [107,109]. The estrogen/ER complex can change the rate of transcription by either binding directly to estrogen-response elements (ERE) or indirectly through interactions with other transcription factors (specificity protein 1, Activator protein 1, etc.) [110,111,112]. Non-genomic signaling has recently emerged as another significant mechanism for estrogen-mediated regulation. Upon activation by estrogens, GPER and likely membrane-associated estrogen receptors activate multiple intracellular signaling pathways, including MAPK, eNOS, and phosphoinositide 3-kinase (PI3K). GPER specifically induces activation of the epidermal growth factor receptor to extracellular signal-regulated kinases [105,113,114,115].

The effects of estrogen vary from receptor to receptor and from tissue to tissue. Multiple examples of this exist, but perhaps the best defined is that of cyclin D1. The regulation of cyclin D1 by estrogens is incredibly complex; it is indirectly transcriptionally regulated, as its promoter contains no ERE. Yet, it is also regulated through PKA and PI3K/protein kinase B (AKT) activity [116,117]. Furthermore, activation of ERα and ERβ demonstrates very different results. Liu et al. have shown specifically that E2/ERα positively regulated cyclin D1 transcription, while E2/ERβ resulted in repression of cyclin D1 transcription and blocked E2/ERα activity [118]. Other examples include vascular smooth muscle, where inducible nitric oxide synthase is positively regulated by ERβ but negatively regulated by Erα [119]. Likewise, using female mice lacking ERs, Jesmin et al. found VEGF to be predominantly regulated by ERα, not Erβ [120].

### 3.1. Vascular Effects of Estrogen

In women, an observable surge in cardiovascular disease occurs around mid-life, especially following the transition after menopause. This observation has led to the understanding that female hormones play a significant role in cardiovascular health, and their loss propels cardiovascular pathology (Table 1). Furthermore, the mechanism of cardiovascular injury in females is more commonly associated with microvascular changes resulting in clinical phenotypes of diastolic dysfunction, hypertension, and coronary microvascular dysfunction [121,122]. Great strides have been made in bringing awareness to the gender differences in cardiovascular disease, and work is ongoing to more specifically define the protective effects of estrogens on cardiovascular health. 

#### 3.1.1. Nitric Oxide

The role of estrogen in intracellular processes is tightly linked to cardiovascular health. Estrogen signaling impacts lipid metabolism, the coagulation system, and antioxidant production, all of which promote the health of the vasculature. Most notably, estrogen is known to be a vasodilator; specifically in the endothelium, estrogen increases the bioavailability of vasoactive molecules, such as nitric oxide and prostaglandins [123]. Non-genomic signaling between ERα and the PI3K-Akt pathway rapidly induces nitric oxide production through phosphorylation of eNOS [124]. Estrogen also regulates eNOS availability through classical genomic mechanisms, resulting in estrogen-dependent control of a rapid and long-term vasodilatory effect [125]. Conversely, estrogen inhibits the synthesis of the potent vasoconstrictor, endothelin-1, through both a direct ERα mechanism and an indirect estrogen metabolite interaction [126]. 

One of the major changes in endothelial function that occurs with the loss of estrogen is the reduction of nitric oxide. Not only is there a reduction in the genomic regulation of eNOS expression, but studies have also reported both a decrease in L-arginine, an eNOS precursor, and increased eNOS inactivation due to increased ROS production [127,128]. This global reduction in nitric oxide leads to reduction of the vasodilatory response, including flow-mediated vasodilation, and to an increase in vasospasm in ischemic vascular beds [129].

#### 3.1.2. Inflammation

Contrary to their role in the immune system, estrogens are also known to have a significant anti-inflammatory role in the cardiovascular system. In rat cerebral arteries, a 50% reduction in superoxide production via NADPH oxidase was shown in females compared to males, likely due to NADPH oxidase activity [130]. Estrogen also regulates pro-inflammatory cytokine production, with estrogen-deficient mice expressing high levels of TNFα and subsequently demonstrating vascular dysfunction [131]. Furthermore, in endothelial cells, estrogen attenuated ROS-induced cytochrome c release and apoptosis, again demonstrating its multifaceted cardiovascular protective effects [132]. Loss of estrogen is associated with inflammatory signaling, with increases in multiple inflammatory cytokines, such as TNFα, IL-1β, and IL-6 [133]. Estrogen deficiency also contributes to plaque formation through changes in adhesion molecules, intercellular adhesion molecule-1 (ICAM-1) and platelet endothelial cell adhesion molecule-1 (PECAM-1), and to vascular remodeling with leukocyte invasion [134]. The overall response to this estrogen-deficient vasculature is that of a pro-inflammatory, stiffened blood vessel with reduced adaptability to environmental stimuli.

**Table 1 ijms-26-02105-t001:** Effects of estrogen in the vasculature.

Activity	Microvascular Effect	Estrogen Source	Reference
Platelet Function	Reduced platelet aggregation	Women on HRT	[134]
Thrombosis	Reduced thromboembolic events	Mice with chronic estradiol exposure	[132]
Inflammation	Reduced NADPH and superoxide production	Female rats	[129]
Reactive Oxygen Species	Increased levels of inflammatory cytokines	Estrogen-deficient animal models	[127,129]
Nitric Oxide Synthesis	Vasodilation through increased eNOS	Estrogen-deficient ewes with estradiol exposure and in vitro models	[120,121]
Adhesion	Increased ICAM-1 and PECAM-1 activity	Estrogen-deficient mice with estradiol exposure	[130]

#### 3.1.3. Thrombosis

Thrombosis risk in relation to estrogen levels is complex, as exogenous hormones, especially in premenopausal women, increase the risk of venous thrombosis, while postmenopausal women with a lack of estrogen are also known to have a greater thrombosis risk, including that of stroke, myocardial infarction, and pulmonary embolism [135]. Estrogen is known to modulate the coagulation profile through transcriptional regulation of hepatic coagulation factor production including factors VII and XII. Genome-wide studies have also identified EREs in a multitude of human coagulation proteins [136,137]. The procoagulant state for patients taking oral contraceptives is especially pronounced with an increase in factors VII, VIII, and fibrinogen and a decrease in the anticoagulant, protein S. 

With more targeted investigations, the type of estrogen and route of administration have also been found to play a significant role in the risk of thrombosis. For example, ethinylestradiol, a synthetic estrogen found in oral contraceptives, is especially notable for increasing the procoagulant effect, due to its slow metabolism and possibly first-pass effects. In several randomized control trials, synthetic estradiol has been found to provide a significantly reduced risk of procoagulant factor production compared to ethinylestradiol [138,139,140].

The increase in thrombosis that occurs with aging and estrogen deficiency may in part be due to increases in inflammatory pathways and loss of nitric oxide, both of which drive platelet activation and adhesion of both the venous and arterial systems. The effects of estrogen on platelet function are controversial. In a mouse model, chronic exposure to exogenous estradiol reduced platelet aggregation and carotid artery thrombosis [141]. Direct treatment of platelets with estradiol can potentiate platelet aggregation through integrin GPαIIb/β3 activation, yet patients on estrogen replacement demonstrate an estrogen-dependent modulation of calcium/cAMP signaling with decreased platelet aggregation [142,143]. 

Fluctuations in hormone levels that occur with the menstrual cycle are associated with variations in several platelet properties, including adhesion and reactivity. Additionally, while not assessing estrogen directly, older females have been shown to have increased platelet reactivity when compared to age-matched men, although this effect was mitigated by aspirin therapy and is yet to be shown to be clinically relevant [144,145]. While more studies need to be completed, the relationship between estrogen and thrombosis is likely more complex than related to platelet activation alone. 

## 4. Estrogen, Purinergic Signaling, and MVD

### 4.1. Estrogen and MVD

Multiple conditions that affect women more commonly than men are also highly associated with the lifetime risk of MVD. These disease entities are wide-ranging, including preeclampsia, collagen vascular disorders such as systemic lupus erythematosus or systemic sclerosis, and lastly Alzheimer’s disease. Of further note, rates of MVD in postmenopausal women are also greater than for age-matched men in other conditions such as diabetes mellitus and hypertension, where there are closer matches in gender prevalence or even male predominance [146]. While the development of MVD is multifactorial, the higher frequency in women and the increases in MVD diagnosis following menopause suggest causative roles for the development of estrogen deficiency. 

Despite the correlation between female sex and incidence of MVD, the specific role of estrogen remains elusive. Most studies addressing this question have focused on the vasodilatory effect of estrogen through modulation of flow-mediated relaxation, calcium channel activity, or NO production or, conversely, through the inhibition of the vasoconstrictor, endothelin-1 [147,148,149,150]. However, aging alone is also shown to increase arterial resistance in vitro [151]. In coronary MVD specifically, estrogen is protective against the development of MVD, through not only vasodilation but also stimulation of angiogenesis and reduced fibrosis [6]. In ovariectomized rabbits, coronary vascular reserve, a marker of microvascular dysfunction, is reduced, with improvement to baseline levels following E2 supplementation [152]. However, the restorative effect of supplemental E2 in human females has not been demonstrated, despite multiple studies addressing the question [153]. These findings suggest that the mechanism of MVD is more complicated than chronic vasoconstriction alone and this necessitates a comprehensive approach to treatment, especially for postmenopausal females.

### 4.2. Purinergic Signaling and MVD

Pathologic stimuli including ischemia/hypoxia, inflammation, and excessive pressure gradients, all contribute to MVD. While much is known regarding the relationship between purinergic pathway activity, endothelial health, and these injurious conditions, the relationship between them as they relate specifically to coronary MVD is less well described. However, derangements in ATP release and signaling in animal models have been linked with systemic diseases such as diabetes mellitus that result in coronary MVD. 

At the microvascular level, regulation of vascular health is mediated through direct local interactions between red blood cells (RBCs), endothelial cells, and ATP receptors. Changes in sheer stress, vessel deformation or injury, and/or decreased oxygen tension induce ATP release through RBC pannexin-1 channels [154]. Localized ATP then activates the P2Y receptors on adjacent endothelial cells, resulting in increased NO and prostaglandin production as well as localized vasodilation [155]. RBC-mediated ATP release and accompanying purinergic regulation can be impaired at times of chronic inflammation, triggering a cascade of injury, with resultant microvascular dysfunction [156]. 

Vasodilation occurs through cell—cell ATP-dependent purinergic communication from the capillaries to the arterioles via connexin and pannexin channel signaling [157,158]. With chronic inflammation and injury, altered connexin and pannexin activity occurs, leading to changes in ATP-release and responses. Several murine connexin knock-out models have demonstrated microvascular dysfunction, including the inability to modulate vascular tone and pathologic angiogenesis, following ischemic injury [159,160]. 

Notably, pharmacologic inhibition of connexin activity reduced microvascular permeability and local edema, suggesting modulatory roles for connexins in endothelial barrier function, possibly through the regulation of eATP [161,162]. Similarly, in a mouse model of diabetic retinopathy, decreases in connexin43 hemichannel-mediated ATP release with Tonabersat markedly reduced ATP-mediated inflammasome activation, thus preventing the associated microvascular changes associated with the disease [163]. 

These protective or injurious microvascular responses are likely impacted by the type of insult and duration of activation of ATP release. While the physiologic response of increases in eATP release through these channels is initially protective, the ongoing ATP release that occurs during pathological, chronic inflammation leads to endothelial damage and resultant MVD, as is commonly associated with systemic inflammatory conditions such as diabetes mellitus [164].

### 4.3. Relationship Between Estrogen and Purinergic Signaling

The effects of estrogen and purines on the vasculature are not well defined. Several groups have focused on the relationship between estrogens, purinergic signaling, and neurovascular units. Using ovariectomized mice, ADP-dependent cerebral vasodilation was found to be partially mediated by estrogen, likely through nitric oxide metabolism [165]. Moreover, adenosine, known to be protective in the vasculature, does not demonstrate gender-specific differential expression. However, adenosine metabolism may have a gender predisposition; in the aortas of female mice, levels of ADA are reduced compared to male mice, suggesting increased adenosine as a possible gender-specific vascular protective molecule [166]. 

Estrogen regulates the expression of purinergic pathway proteins in the endothelium (Figure 5). For example, in lung endothelium, estradiol upregulated expression of the A_2A_ receptor, which improved angiogenic and wound healing end points [167]. Likewise, our group has also established a relationship between estrogen, purinergic signaling, and endothelial health. In a recent study, we demonstrated that estradiol upregulated CD39 expression, with subsequent improvement in endothelial wound healing and angiogenesis. Additionally, expression of CD39 was reduced in ovariectomized mice, accompanied by a significant reduction in adenosine, suggesting estradiol to be critical in the maintenance of balance between ATP and adenosine in the cardiovascular system [168]. 

As previously discussed, a highly complex relationship exists between estrogens and thrombotic risk, with wide-ranging results on estrogen levels and platelet activity [169]. Purinergic signaling, specifically, maintains antithrombotic balance, primarily through the regulation of ATP/ADP availability. Purinergic pathway proteins are present on platelets, in particular, CD39 and P2Y_1_, and are more highly expressed on the platelets of female rats as compared to males. Female rat platelets also demonstrated reduced reactivity in response to ADP stimulation [170]. Ovariectomized rats exhibit a significant decrease in hydrolysis of ATP/ADP, resulting in increased platelet aggregation, with estrogen supplementation not altering the outcome [171]. Conflicting results have been shown from multiple studies evaluating the role of HRT on platelet function in postmenopausal women, and as such, understanding of the relationship between estrogens and platelet functionality would benefit greatly from additional studies. 

The association between estrogen and ATP receptors is also complex, with the P2X receptor activity acting in a protective (e.g., osteoporosis) or unfavorable way (e.g., neuronal injury) [172]. Specifically, estrogen has been shown to modulate P2X3 expression in neurons, thus inhibiting the ATP-mediated response to peripheral pain stimulation [173]. Additionally, treatment with estradiol inhibits the cation current of the P2X7 receptor [174]. Furthermore, using a cerebral ischemia model, Thakkar et al. demonstrated that estradiol significantly reduced P2X7 expression, as well as downstream NLRP3 inflammasome signaling [175]. 

Overall, these findings suggest the likelihood of an estrogen-dependent ATP receptor response to inflammatory signaling; however, the specific relationship with microvascular dysfunction is yet to be determined. We propose the hypothesis that, in the microvasculature, protective estrogen-dependent upregulation of CD39 scavenges pathologic concentrations of eATP to ultimately generate adenosine, thereby modulating purinergic-dependent inflammatory effects. The overall effects of vascular CD39 expression are the amelioration of pro-inflammatory, pro-thrombotic effects, as well as improved vasodilation and mitigation of those insults to the microvasculature that culminate in MVD.

## 5. Therapeutic Targets 

Aside from the diagnostic challenges of MVD (see Taqueti and Carli, 2018 for a comprehensive review), targeted treatment for it has also proven difficult [1]. No specific therapies currently exist, with current treatment focused on traditional reduction of cardiovascular risk. Patient-focused strategies, including smoking cessation, weight loss, and regular exercise, along with control of blood pressure, lipids, and glucose management, are likely to reduce overall cardiovascular risk. However, this approach can be very challenging for perimenopausal women, as fluctuating hormones significantly contribute to weight gain, insulin resistance, and adiposity [176]. There remains an ongoing need for additional, targeted therapies for patients with MVD to prevent morbidity and mortality.

The use of exogenous estrogen has fluctuated over the past decades, as several large clinical trials have demonstrated risks associated with its long-term use. Several landmark studies, including the WHI, have directed the use (or lack of use) of exogenous estrogens for hormonal replacement therapy. The WHI, one of the largest studies ever enrolled, assessed the use of estrogen and estrogen plus progesterone in aging women [177]. The predominant form of estrogen replacement was oral, with a small subset using local estrogen supplementation. These studies were stopped early, due to concerns about increased risk of stroke, thrombosis, dementia, and some cancers, including endometrial and breast. While the original results have begun to be reconsidered, especially concerning the risk of breast cancer, there remains a substantial risk associated with long-term use of exogenous estrogen [177,178].

Notably, the use of estrogens in patients greater than sixty years old or more than ten years post-menopause is associated with an increased risk of stroke, thrombosis, and possibly dementia. An assessment of the WHI hormone therapy trial findings, including post-intervention follow up, describe a significant risk of stroke and venous thrombosis, regardless of age of HRT initiation [177]. Similarly, a nationwide cohort study from Korea also found an increased risk of stroke in postmenopausal women taking HRT [179]. A recent meta-analysis of 33 randomized controlled trials investigating the use of HRT on cardiovascular outcomes also demonstrated an overall increase in ischemic stroke and thrombosis risk [180]. The risk of dementia in relation to HRT use is poorly understood, as multiple observational studies have generated conflicting results [181,182]. However, the more recent Kronos Early Estrogen Prevention Cognitive and Affective Study (KEEPS) continuation study did not find an association at ten years follow-up between prior HRT use and cognitive decline [183].

Importantly, the benefits of taking long-term estrogen replacement have not been shown to significantly reduce the risk of cardiovascular disease in older females, although investigation of this question would also benefit from additional randomized control trials [184]. A paucity of data exist for the use of exogenous estrogen, especially for non-menopausal females (e.g., transgender patients), with a multitude of questions remaining about the use of estrogens, including time of initiation, dosing, type of synthetic estrogen, route of administration, and duration of use. Before further characterization of these questions, wide-scale use of estrogens in patients greater than sixty years old is unlikely to occur. Current recommendations for HRT for vasomotor menopause symptoms or menopausal hormone therapy are limited to perimenopausal women younger than sixty, with use restricted to less than ten years. A subset of women with elevated risk (prior stroke, breast cancer, etc.) are not eligible for use at all. 

The lack of specificity for MVD and the complexity of estrogen replacement therapy suggest that a novel approach to the treatment of MVD is warranted. The purinergic signaling pathway is emerging as a likely therapeutic avenue for the treatment of MVD. Various targets in the purinergic signaling pathway are known to be estrogen-dependent and are currently under investigation in a variety of conditions, including cancer, cardiovascular disease, and neurological disorders. 

### 5.1. CD39 Modulation

Promotion of adenosine production from ATP hydrolysis is being considered in various diseases, especially those requiring targeting of inflammation and immunomodulation. For example, in an inflammatory bowel disease model, both enzymatic hydrolysis of ATP by a recombinant apyrase (APT102) and overexpression of CD39 reduced inflammation and colitis [185]. Another approach is through CD39-meditated inhibition of eATP activity. Recently, it was demonstrated that a soluble, recombinant CD39 was able to inhibit ATP or ADP activity. This reaction resulted in the production of AMP and offers the promise of a novel approach to pharmacologic control of purinergic activity [186]. Most of the work on CD39 modulation lies within cancer therapeutics, as excessive adenosine is primarily immunosuppressive. CD39 antagonism with either chemical inhibitors or antibodies has shown promise in reducing adenosine and promoting an ATP-driven, pro-inflammatory, cytotoxic, anti-tumor environment [187]. This environment, however, would not be supportive of a healthy microvasculature, as a strong anti-angiogenic effect has been observed with anti-CD39 therapies [188]. It remains to be seen whether long-term use of these therapies will have a significant effect on the microvascular health of aging patients. 

### 5.2. Purine Modulation

Targeting purine production and availability has also been pursued as a mechanism of interest. Dipyridamole is a classic example of this. Dipyridamole is a nucleoside transporter and phosphodiesterase inhibitor whose use is known to increase interstitial adenosine as well as intracellular cAMP and cGMP. It has traditionally been used to reduce platelet aggregation and platelet-related thrombosis, as well as to act as a vasodilator, but was largely replaced with more advanced antiplatelet and vasodilatory agents [189]. Its use currently is limited to cardiac stress testing. However, it has the potential to be reconsidered for use in microvascular disease. 

Methotrexate (MTX) is another older medication that affects purine metabolism. Frequently used for inflammatory autoimmune conditions such as rheumatoid arthritis, vasculitis, and inflammatory bowel disease, MTX is an anti-metabolite that inhibits the enzyme dihydrofolate reductase, thus reducing nucleotide synthesis. A different mechanism is understood for the effect of MTX in autoimmune disease; adenosine deamination by ADA is reduced due to MTX inhibition of AICAR transformylase. Thus, patients taking MTX have increased concentrations of extracellular adenosine and increased adenosine-induced vasodilation [190]. MTX use is associated with a reduced risk of mortality from cardiovascular disease in patients with rheumatoid arthritis [191]. Like dipyridamole, MTX has the potential for use in MVD. Further trials including risk–benefit analyses are needed to investigate the repurposing of these medications for benefit in MVD. 

### 5.3. Purine Receptors

#### 5.3.1. Adenosine Receptors 

Adenosine has been used clinically for the diagnosis and termination of supraventricular tachycardias; however, its effect is short-lived and non-specific. In recent years, specific modulation of the adenosine receptors has been undertaken with multiple compounds that are in various stages of clinical development. While numerous compounds have been synthesized, many have not proceeded further than clinical trials, due to either lack of efficacy or difficulty with administration. Additionally, systemic administration of A_2A/2B_ receptor agonists has the untoward side effect of hypotension. 

Most of the progress in A_2A_ receptor-specific agonist development has focused on compounds used for myocardial perfusion imaging. For example, regadenoson and binodenoson are designed for coronary vasodilation, with regadenoson currently in clinical use for perfusion scans [192]. More recently, regadenoson and another A_2A_ receptor-specific agonist, ATL1223, were shown to increase survival, with reduced neurological injury in pigs following cardiac arrest and extracorporeal membrane oxygenation, possibly due to a decrease in inflammation [193]. Additional preclinical studies have also demonstrated the anti-inflammatory and vasodilatory effects of A_2A_ receptor activation of various compounds. For example, CGS21680 is shown to increase cAMP in vascular smooth muscle and to reduce neuropathic pain through anti-inflammatory signaling [194,195]. Endothelial dysfunction has also been examined using a different compound, PSB0777, which restored endothelial barrier function, vasodilation, and adhesion in cirrhotic rats [196,197,198]. Similarly, in rats, CGS21680 demonstrated stimulation of eNOS in renal afferent arterioles. While CGS21680 is no longer in clinical development due to hemodynamic outcomes, it is a clear example of the possibility of effects that can be achieved through selective adenosine receptor activation. The A_2B_ receptor is a low-affinity receptor, and the development of specific sub-type agonists has been limited. One study, however, demonstrated, using BAY 60-6583, an A_2B_ receptor selective agonist, regional selectivity to the vasodilatory response, specifically vasodilation in the renal and mesenteric vascular beds, but not in the skeletal muscle [198]. 

Although primarily protective in the vasculature, adenosine receptors are implemented in the progression of various diseases, including neurodegenerative disorders, asthma, and cancers. Because of this, multiple adenosine receptor antagonists are also in development. For example, the A_2A_ receptor antagonist KW-6002 (istradefylline) is currently in use as an adjunct medication for Parkinson’s disease, with others in development [199]. Furthermore, activation of these receptors has been associated with the progression of some cancers through the promotion of immune tolerance. In summary, while the concept of an adenosine receptor agonist is promising, the complexity of not only receptor but also tissue specificity and targeting remains to be resolved. 

#### 5.3.2. ATP Receptor Antagonists

Of the fifteen receptor subtypes for ATP, approximately thirteen are present in the cardiovascular system from both the inotropic P2Y and the metabotropic P2X receptor families. Multiple compounds aimed at blocking the actions of various P2Y and P2X ATP receptors are under development. The most well-known therapeutics are thienopyridines (clopidogrel, ticlopidine, and prasugrel), a class of drugs that prevents platelet thrombosis via inhibition of the P2Y_12_ receptor [200]. Since its initial identification as a platelet receptor, P2Y_12_ has also been isolated in microglia, vascular smooth muscle, and eosinophils. While primarily used for antiplatelet therapy, recent studies have shown thienopyridine-induced alterations in inflammatory responses. In a sepsis model of inflammation, Liverani et al. showed inhibition of P2Y_12_ via knock-out mice or pharmacologically reduced platelet–leukocyte interactions and overall inflammation [201]. Other researchers have begun investigating the role of thienopyridines in other inflammatory conditions, such as lung injury and cancer [202]. Studies are limited with respect to thienopyridines, P2Y_12_ inhibition, and estrogen; however, it was recently demonstrated that mice treated with clopidogrel were protected from bone loss associated with estrogen deficiency, suggesting a protective role of ATP receptor antagonism in conditions related to lack of estrogen, such as MVD [203]. 

Newer compounds targeting additional ATP receptors, including P2X3, P2X4, and P2X7, are also emerging. Studies specific to the microvasculature are limited but are beginning to show potential. In a rat model of chronic hypertension, vascular resistance was decreased in renal arterioles following exposure to a nonselective P2X receptor antagonist (pyridoxal phosphate-6-azo (benzene-2,4-disulfonic acid) tetrasodium salt hydrate (PPADS)), likely due to increased nitric oxide production [204]. Additionally, endothelial integrity is a target of interest for ATP receptor antagonists, with multiple studies demonstrating improvement in barrier function and reduced edema with antagonism of several P2 receptors, including P2X4, P2X7, and P2Y_1_ [205]. 

Recent studies have also investigated the regulation of the ATP- and UTP-mediated inflammatory response in the vasculature. For example, pharmacologic inhibition of P2Y_6_ using MRS 2578 in mice attenuated inflammatory responses [206]. The P2X7 receptor, specifically, has been extensively studied in relation to its role in inflammation. Multiple compounds are in development phases for conditions such as neuroinflammatory disorders, rheumatoid arthritis, and inflammatory bowel disease. While several compounds (AZD9056 and CE-224,535) have failed phase II testing, these endeavors signify an ongoing interest in this pathway as a novel modulator of the inflammatory response [207]. Reduction in the detrimental inflammatory cascade would be essential to the prevention of microvascular injury. P2 receptors have great potential for the regulation of vascular resistance, inflammation, and endothelial integrity; however, additional studies directed specifically at the microvasculature response are still needed. 

## 6. Conclusions and Future Perspectives

Microvascular disease remains a challenge, both for diagnostics and effective treatments. The observation of MVD occurring disproportionally in females is highly suggestive of the role of an estrogen-dependent protective mechanism. Loss of estrogen exposes patients to vasoconstriction, inflammation, and injury from shear stress, all of which contribute to microvascular dysfunction. Purines are also protective in the microvasculature, specifically through vasodilation and anti-inflammatory pathways. Better defining relationships between estrogen and purinergic signaling will expand the possibilities for the development of therapies to treat MVD. Identification of novel targets or repurposing existing medications both have the potential to be innovative treatments for MVD. Further characterizing the relationships between estrogen, purinergic signaling, and MVD will only expand the prospective therapeutic targets in the vasculature and putatively into other inflammatory conditions. Continued research in this field can potentially improve vasodilation, reduce the risk of thrombosis, develop targeted anti-inflammatory therapies, and ultimately enhance clinical outcomes in patients with MVD.

## Figures and Tables

**Figure 1 ijms-26-02105-f001:**
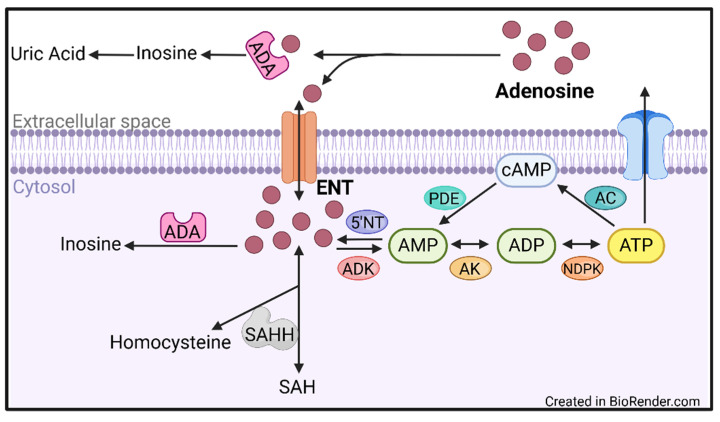
Adenosine production and metabolism. Extracellular adenosine is rapidly deaminated to inosine or shuttled intracellularly via ENT transmembrane channels. Intracellular adenosine is either deaminated to inosine via adenosine deaminase (ADA), recycled in the purine salvage pathway, or re-phosphorylated to ATP. 5′NT: 5′-nucleotidase, AC: adenylate cyclase, ADA: adenosine deaminase, ADP: adenosine diphosphate, ADK: adenosine kinase, AK: adenylate kinase, AMP: adenosine monophosphate, ATP: adenosine 5′-triphosphate, cAMP: cyclic adenosine monophosphate, ENT: equilibrative nucleoside transporter, NDPK: nucleoside-diphosphate kinase: PDE; phosphdiesterase; SAHH: S-adenosyl homocysteine hydrolase.

**Figure 2 ijms-26-02105-f002:**
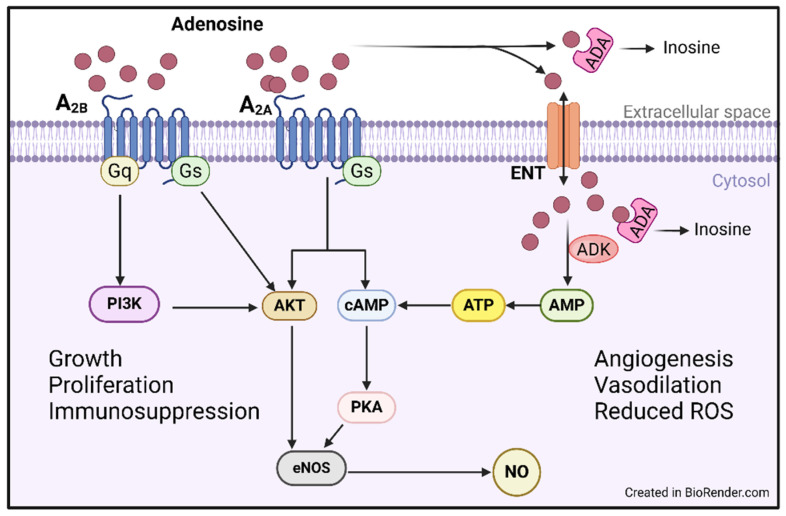
Adenosine activity in endothelial cells. Extracellular adenosine interacts with endothelial cells in several ways resulting in vasoprotection. 1. Adenosine activates ADORA receptors on the cell surface stimulating activation of pathways including PI3K and PKA signaling. 2. Adenosine is transported intracellularly through ENT channels where it can be deaminated to inosine or utilized in further downstream signaling. ADA: adenosine deaminase, ADK: adenosine kinase, AKT: protein kinase B, AMP: adenosine monophosphate, ATP: adenosine 5′-triphosphate, cAMP: cyclic adenosine monophosphate, eNOS: endothelial nitric oxide synthase, ENT: equilibrative nucleoside transporter, NO: nitric oxide, PKA: protein kinase A, PI3K: phosphatidylinositol-3 kinase.

**Figure 3 ijms-26-02105-f003:**
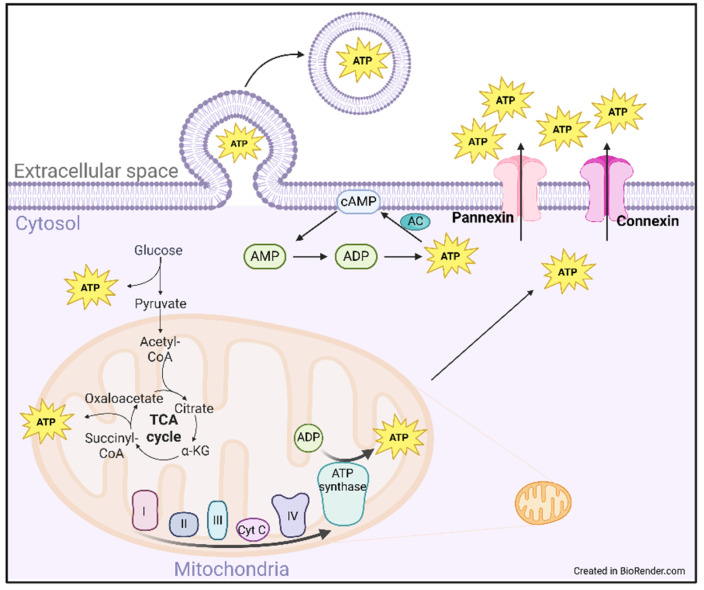
ATP production and transport. ATP is produced 1. through the metabolism of glucose (glycolysis), 2. through the metabolism of pyruvate in the tricarboxylic acid (TCA) cycle, and 3. through oxidative phosphorylation in the electron transport chain. Once intracellular, ATP is released via transmembrane transporter channels, pannexins and connexins. Intracellular ATP is also released though controlled exocytosis. AC: adenylate cyclase, ADP: adenosine diphosphate, AMP: adenosine monophosphate, ATP: adenosine 5′-triphosphate, cAMP: cyclic adenosine monophosphate, Cyt C: cytochrome C.

**Figure 4 ijms-26-02105-f004:**
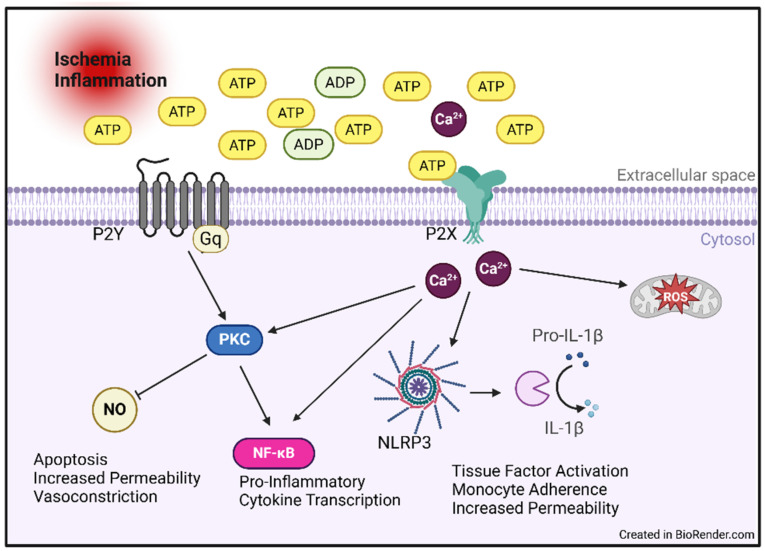
ATP pathway activation in endothelial cells. Extracellular ATP is present due to uncontrolled release following inflammatory and/or ischemic events. ATP stimulates endothelial cells through activation of several cell surface receptors. 1. ATP activates P2Y receptors on the cell surface stimulating downstream activation of PKC signaling, including inhibition of NO (noted by blunt arrowhead). 2. ATP activates P2X receptors on cell surface leading to calcium release and increased NLRP3 inflammasome and ROS activity. ADP: adenosine diphosphate, ATP: adenosine 5′-triphosphate, IL-1β: interleukin-1β, NF-κB: nuclear factor-kappa B, NLRP3: nucleotide-binding domain, leucine-rich–containing family, pyrin domain–containing-3, NO: nitric oxide, PKC: protein kinase C, P2X: purinergic P2X receptor, P2Y: purinergic P2Y receptor, ROS: reactive oxygen species.

**Figure 5 ijms-26-02105-f005:**
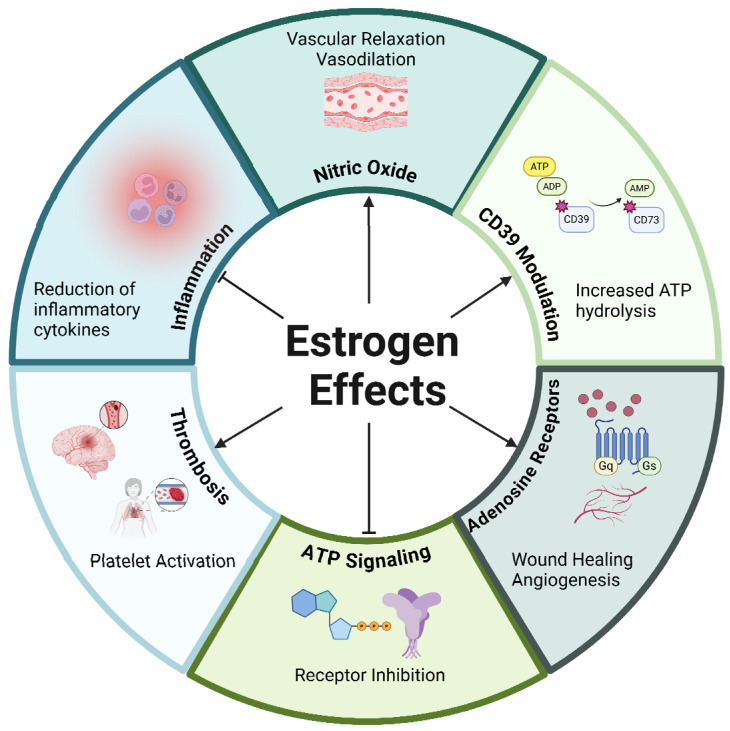
Effects of estrogen on vascular function: The role of estrogen on endothelial processes with a focus on purinergic-specific pathways. ↑ denotes a positive effect, ⊥ denotes an inhibitory effect. ADP: adenosine diphosphate, ATP: adenosine 5′-triphosphate, CD39: ectonucleoside triphosphate diphosphohydrolase-1 (ENTPD1), CD73: ecto-5′-nucleotidase (NT5E).

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
