# Peer review of "Impact of Estrogen on Purinergic Signaling in Microvascular Disease"

_ijms, 2025, doi:10.3390/ijms26052105_

Round 1

Reviewer 1 Report

Comments and Suggestions for Authors

In this review manuscript, the authors aim to explore the relationship between purinergic and estrogen signaling and MVD. While previous reviews have discussed MVD, the authors' perspective—focusing on purinergic signaling—is novel, as there is no existing literature on this specific angle. I have several suggestions for improvement or clarification.

Major Suggestions:

  1. The manuscript introduces purinergic and estrogen signaling separately. However, I find that the causal relationship between these signaling pathways and MVD is not directly addressed. For example, the authors first describe various purinergic signals (adenosine, ATP, etc.) and then discuss how environmental stressors such as inflammation and hypoxia affect purinergic signaling. My question is: Is there direct evidence showing that purinergic signaling is altered in MVD patients? For instance, what is the expression pattern of connexins in MVD patients?
  2. Similar to the first point: Are there any reports indicating abnormal estrogen signaling specifically in MVD patients (excluding indirect evidence from inflammation or other factors)?
  3. When outlining the relationship between purinergic and estrogen signaling in the context of MVD, the manuscript should provide more explanation regarding the factors that induce MVD. Although the authors state in lines 517–519 that there is no existing literature on this topic, I suggest they attempt to propose a plausible causal hypothesis, as this is a key focus of the manuscript.

Minor Suggestions:

  1. The order of figure and text descriptions is inconsistent. For example, Figure 2 is described in line 284, but Figures 3–4 are already discussed earlier.
  2. In line 283, the authors mention P1 receptors A2A and A2B. However, in coronary MCD, the A1 receptor is also important and should be discussed as well.

Author Response

Please see attached responses. 

Reviewer 2 Report

Comments and Suggestions for Authors

This review explores the interplay of purinergic and estrogenic signaling in microvascular disease. It covers in detail the dual roles of these pathways in protective and pathological cardiovascular processes, with a particular focus on post-menopausal women and transgender individuals undergoing hormone replacement therapy. The manuscript underscores the ongoing need for continued research into the crosstalk between these pathways in order to develop targeted therapies to improve cardiovascular health with reduced adverse events.

The paper presents comprehensive and extensive coverage of the most current relevant literature, and cogent discussions are provided throughout. It does a fine job addressing an important knowledge gap by linking the two pathways, with significant implications for cardiovascular health. It is well-written and easy to follow, with appropriate tables and figures to guide understanding for the reader.

There is only one minor apparent omission that I recommend be addressed. Regarding estrogenic signaling and thrombosis, there is extensive literature pointing to roles for estrogen in regulation of expression of coagulation factors in the liver. These findings comprise a key part of the current understanding of roles of estrogen in thrombosis. There is a passing reference to this in section 3.1.1 dedicated to nitric oxide signaling, but a brief, expanded review and some discussion on this area would round out this manuscript.

Author Response

Please see attached responses. 

Round 2

Reviewer 1 Report

Comments and Suggestions for Authors

Thank you for your email. I have carefully reviewed the author's response letter and compared it with the revised manuscript. The author has indeed addressed all of my suggestions, and I have no further comments. Therefore, I recommend that the revised manuscript be accepted for publication in IJMS. Thank you.